# Fabrication and Evaluation of Alginate/Bacterial Cellulose Nanocrystals–Chitosan–Gelatin Composite Scaffolds

**DOI:** 10.3390/molecules26165003

**Published:** 2021-08-18

**Authors:** Zhengyue Li, Xiuqiong Chen, Chaoling Bao, Chang Liu, Chunyang Liu, Dongze Li, Huiqiong Yan, Qiang Lin

**Affiliations:** 1Key Laboratory of Tropical Medicinal Resource Chemistry of Ministry of Education, College of Chemistry and Chemical Engineering, Hainan Normal University, Haikou 571158, China; zhengyedu@163.com (Z.L.); chenxiuqiongedu@163.com (X.C.); 2Key Laboratory of Natural Polymer Functional Material of Haikou City, College of Chemistry and Chemical Engineering, Hainan Normal University, Haikou 571158, China; bcl0409@163.com (C.B.); liuchang202107@126.com (C.L.); liuchunyang212021@126.com (C.L.); dongzeli2019@163.com (D.L.); 3Key Laboratory of Water Pollution Treatment & Resource Reuse of Hainan Province, College of Chemistry and Chemical Engineering, Hainan Normal University, Haikou 571158, China

**Keywords:** alginate, bacterial cellulose nanocrystals, internal gelation, layer-by-layer electrostatic assembly, composite scaffolds, bone tissue engineering

## Abstract

It is common knowledge that pure alginate hydrogel is more likely to have weak mechanical strength, a lack of cell recognition sites, extensive swelling and uncontrolled degradation, and thus be unable to satisfy the demands of the ideal scaffold. To address these problems, we attempted to fabricate alginate/bacterial cellulose nanocrystals-chitosan-gelatin (Alg/BCNs-CS-GT) composite scaffolds using the combined method involving the incorporation of BCNs in the alginate matrix, internal gelation through the hydroxyapatite-d-glucono-δ-lactone (HAP-GDL) complex, and layer-by-layer (LBL) electrostatic assembly of polyelectrolytes. Meanwhile, the effect of various contents of BCNs on the scaffold morphology, porosity, mechanical properties, and swelling and degradation behavior was investigated. The experimental results showed that the fabricated Alg/BCNs-CS-GT composite scaffolds exhibited regular 3D morphologies and well-developed pore structures. With the increase in BCNs content, the pore size of Alg/BCNs-CS-GT composite scaffolds was gradually reduced from 200 μm to 70 μm. Furthermore, BCNs were fully embedded in the alginate matrix through the intermolecular hydrogen bond with alginate. Moreover, the addition of BCNs could effectively control the swelling and biodegradation of the Alg/BCNs-CS-GT composite scaffolds. Furthermore, the in vitro cytotoxicity studies indicated that the porous fiber network of BCNs could fully mimic the extracellular matrix structure, which promoted the adhesion and spreading of MG63 cells and MC3T3-E1 cells on the Alg/BCNs-CS-GT composite scaffolds. In addition, these cells could grow in the 3D-porous structure of composite scaffolds, which exhibited good proliferative viability. Based on the effect of BCNs on the cytocompatibility of composite scaffolds, the optimum BCNs content for the Alg/BCNs-CS-GT composite scaffolds was 0.2% (*w*/*v*). On the basis of good merits, such as regular 3D morphology, well-developed pore structure, controlled swelling and biodegradation behavior, and good cytocompatibility, the Alg/BCNs-CS-GT composite scaffolds may exhibit great potential as the ideal scaffold in the bone tissue engineering field.

## 1. Introduction

Bone tissue engineering is one of the most promising approaches to achieve bone repair and regeneration, making use of a combination of cells, engineering and materials together with suitable biochemical and physicochemical factors [1,2]. The ideal tissue engineering scaffold needs to be biocompatible, biodegradable and processable, have sufficient mechanical strength to withstand physiological stress, and have three-dimensional (3D) porous network structures that closely mimic the natural extracellular matrix (ECM), and thus be able to support mineral matrix deposition as well as cell attachment, proliferation and differentiation [3,4].

Since natural polymers offer very similar advantages to natural ECM, these ECM-like polymers, such as alginate, chitosan, gelatin, etc., have been widely used as biomaterials for the fabrication of tissue engineering scaffolds [5,6]. Alginate is a natural anionic polysaccharide extracted from brown algae and various bacteria [7]. Its heteropolysaccharide backbone comprises (1–4)-linked β-d-mannuronic acid and α-l-guluronic acid blocks distributed in an irregular blockwise pattern [8]. Alginate can be easily crosslinked by ionic interactions between carboxylic acids and divalent cations, usually calcium, to form stable hydrogel [9]. Due to its excellent biocompatibility, biodegradability, non-immunogenicity and 3D polymeric network structure, alginate hydrogel has been commonly used as the scaffold material for tissue engineering [10].

In spite of its favorable merits, pure alginate hydrogel may be incapable of meeting all the requirements of the ideal scaffold because of its inherent drawbacks. Firstly, the fatal drawback is the lower and unstable mechanical strength, which makes it inappropriate for a load-bearing bone scaffold [11]. Secondly, the potential limitation in using alginate hydrogel in tissue engineering is the lack of cell recognition sites for the cell adhesion and differentiation [12]. Furthermore, the cross-linking structures produced by alginate and the divalent cation, Ca^2+^, that are easily destroyed in biological buffers containing chelators or monovalent electrolytes, result in the extensive swelling and uncontrolled degradation [13]. On the basis of these problems, one effective approach to overcome the mechanical and biological limits of alginate hydrogel is the use of bacterial cellulose nanocrystals (BCNs) as reinforcing components in the alginate matrix [14,15]. BCNs are ideal materials for the fabrication of tissue engineering scaffold owing to their referred biocompatibility, potential low cost, renewable raw material nature and desired mechanical properties [16,17,18,19]. It was reported that low cellulose nanocrystal concentrations effectively improved the mechanical properties of scaffolds without significant cytotoxic effects [20]. In addition, porous cellulose scaffolds and their hybrids with chitosan and alginate were fabricated for various applications in tissue engineering [21]. Another versatile method to improve the mechanical properties of alginate hydrogel is internal gelation, which allows homogeneous alginate hydrogel to be obtained by the controlled release of Ca^2+^ from the insoluble calcium salts, such as hydroxyapatite (HAP), calcium carbonate (CaCO_3_) and calcium sulfate (CaSO_4_), in the presence of d-glucono-δ-lactone (GDL) [1,10,22]. This method avoids the unbalanced crosslinking density, thus enhancing the mechanical properties and homogeneity of the hydrogel. Last but not least, the design of scaffolds must take the interactions between cells and the surfaces of scaffold materials into consideration. It is well known that the Arg–Gly–Asp (RGD) tripeptide found in cell-adhesive proteins such as vitronectin, laminin, fibronectin, collagen or gelatin plays an important role in the control of specific cell behaviors, including adhesion, migration, proliferation and differentiation [23,24,25,26]. Therefore, the coating of the RGD tripeptide on the surface of the scaffold is critical to achieve specific biological functions. The surface modification of the scaffold can be carried out by some anchoring processes such as cross-linking [26,27], benzyl-protected phosphonic anchors [28], thiol and phosphonate anchors [29], dopamine anchoring [30,31], photopolymerization [32] and layer-by-layer assembly [33]. Among these methods, a simple and versatile technique of layer-by-layer (LBL) electrostatic assembly of polyelectrolytes can be applied to improve the mechanical and biological properties of alginate hydrogel. The LBL electrostatic assembly is performed through an alternating deposition of polyelectrolytes via electrostatic interaction on the surface of alginate hydrogel, which may provide it with new functionality [2].

In the present study, on account of the excellent biocompatibility, bioactivity and osteoconduction of HAP [34], HAP-GDL complex was used as the gelling system for the preparation of homogeneous hydrogel. The BCNs, as reinforcement in the alginate matrix, were applied to enhance the porous microstructure, thus helping to attain the desired mechanical and biological activity of the composite scaffolds because of their excellent biodegradability, low cytotoxicity and suitable mechanical properties, which are similar to those of natural tissue. To further regulate biodegradation behavior and facilitate cell attachment and proliferation, successive LBL electrostatic assemblies of chitosan (CS) and gelatin (GT) were conducted on the surface of the composite scaffolds. Finally, the outer GT was further cross-linked by the cross-linking system of 1-Ethyl-3-(3-dimethyl-aminopropyl-1-carbodiimide)/*N*-hydroxysuccinimide (EDC/NHS), which was considered to have low toxicity, thus promoting the stability and mechanical strength of the composite scaffolds [4,35]. The fabrication route of alginate/bacterial cellulose nanocrystals-chitosan-gelatin (Alg/BCNs-CS-GT) composite scaffolds, according to the aforementioned methods, is illustrated in Scheme 1. The effects of various contents of BCNs on the scaffold morphology, porosity, mechanical properties, and swelling and degradation behavior were investigated. Furthermore, the cytotoxicity of the composite scaffolds was also assessed through the use of osteosarcoma MG-63 cells and osteoblastic MC3T3-E1 cells. To the best of our knowledge, the attempted fabrication of Alg/BCNs-CS-GT composite scaffolds using the combined method, involving the incorporation of BCNs in the alginate matrix, internal gelation through the HAP-GDL complex, and LBL electrostatic assembly, with the aim of developing the ideal scaffold for bone tissue engineering, has rarely been reported up to now.

## 2. Results and Discussion

### 2.1. Colloidal and Interfacial Activity of BCNs

It was reported that the sulfuric acid hydrolysis of BC could endow them with anionic sulfate half-ester groups that could be further oxidized by hydrogen peroxide, thus generating BCNs with good colloidal and interfacial activity [36,37]. The SEM image and TEM image of the resultant BCNs are presented in Figure 1. It can be observed that BCNs exhibited short and rod-like microfibrils with very thin layers in their SEM images and bundles or ribbons in their TEM images, due to the removal of the amorphous components and the cleavage of the crystalline microfibrils [38]. The hydrodynamic particle size distribution and zeta potential distribution of BCNs were measured by dynamic light scattering (DLS), with the results shown in Figure 2. Due to the removal of the amorphous region of BC and the cleavage of the glycosidic bond during the hydrolysis of sulfuric acid, the BCNs displayed a narrow particle size distribution with an average hydrodynamic diameter of 317.2 nm (PDI = 0.17), which implied that, in the sulfuric acid hydrolysis of BC, it was easy to obtain fiber crystals of small size. Additionally, it can be seen from Figure 2b that the BCNs were negatively charged because the negatively-charged carboxyl groups had been introduced onto the surface of the BCNs through the sulfuric acid hydrolysis, followed by the hydrogen peroxide oxidation [38]. It is worth noting that the zeta potential of BCNs was at about −36.9 mV, and the absolute value was higher than 30 mV, so they could be stably dispersed in aqueous solution by electrostatic repulsion between the BCNs particles [39]. From the above analysis results, it can be seen that the BCNs possessed a small particle size and good interfacial properties, and therefore, could be used as an ideal filler in the construction of tissue engineering scaffolds.

### 2.2. Morphology of Alg/BCNs-CS-GT Composite Scaffolds

As shown in Figure 3a, the Alg/BCNs-CS-GT composite scaffolds prepared by endogenous cross-linking of the HAP-GDL complex and the LBL assembly of polyelectrolytes, using BCNs as the reinforcing agents, showed regular 3D morphology. Due to the small particle size and good interfacial property of BCNs, they were uniformly dispersed in the alginate matrix, so that the wet Alg/BCNs-CS-GT composite hydrogel were pale blue. Moreover, the composite scaffolds did not shrink, collapse or deform after drying (Figure 3b), but maintained a good morphological structure even during the LBL assembly process (Figure 3c), indicating that the preferred fiber morphology of BCNs was achieved and that their barrier properties could provide physical support for the Alg/BCNs-CS-GT composite scaffolds. After the freeze-drying process, the Alg/BCNS-CS-GT composite scaffolds generated uniform pore structures while maintaining their external 3D architectures, which verified the homogeneous cross-linking of the composite scaffolds. With the increase in BCNs content, the pore structure of the Alg/BCNs-CS-GT composite scaffolds decreased gradually, and the pore size decreased from 200 μm to 70 μm, indicating that the BCNs content had a significant impact on the pore size of the Alg/BCNs-CS-GT composite scaffolds, as shown in Figure 3d–i. It was reported that the pore size of the scaffold material affects the cell adhesion, proliferation and differentiation, and that it must be small enough to support cell growth, but must also meet the requirements to ensure the transfer of metabolic waste and nutrients [40,41]. Therefore, changing the BCNs content to regulate the pore structure of the Alg/BCNs-CS-GT composite scaffolds is of great significance in terms of their cellular behaviors.

In addition, it can be observed from Table 1 that the porosity of the Alg/BCNs-CS-GT composite scaffolds was obviously higher than Alg-CS-GT composite scaffolds, and with the increase in BCNs, the porosity of the composite scaffolds gradually decreased from 86.2% to 73.2%. The above results illustrated that BCNs with fibrillar meshwork structures fully exerted the skeleton effect and maintained the excellent 3D morphology and porous structure of the composite scaffolds.

### 2.3. Characterization of Alg/BCNs-CS-GT Composite Scaffolds

The incorporation of BCNs and the LBL electrostatic assembly of polyelectrolytes may be an effective means to develop composite scaffolds that could fully mimic the strength, stiffness and mechanical behavior of natural tissue in order to withstand physiological loads [1]. The mechanical properties of the composite scaffolds were examined with a universal testing machine. The stress–strain curves of the composite scaffolds are shown in Appendix A, where the highest linear point is the compressive strength of the composite scaffolds.

It can be seen from Figure 4 that the compressive strength of the Alg-CS-GT composite scaffolds was only 0.189 MPa, and the addition of a small amount of BCNs could clearly enhance their mechanical properties, thereby improving the compressive strength of Alg/0.1%BCNs-CS-GT to 0.224 MPa. When the added amount of BCNs exceeded 0.1% (*w*/*v*), the compressive strength of the Alg/BCNs-CS-GT composite scaffolds increased significantly compared to the Alg-CS-GT composite scaffolds. As the content of BCNs increased, their mechanical properties were gradually enhanced, while physical bonds were also established between the components of the system. This result was ascribed to the fiber network structure and interface interaction of BCNs that decreased the pore size of the Alg/BCNs-CS-GT composite scaffolds, thereby resulting in the enhancement of their mechanical properties. Moreover, the filling effect induced by the excellent barrier properties of BCNs also improved the mechanical strength of the Alg/BCNs-CS-GT composite scaffolds. Although the compressive strength of the Alg/BCNs-CS-GT composite scaffolds was indeed enhanced through the supplementation of BCNs from 0.189 MPa to 0.318 MPa, they were not able to provide the stable mechanical strength needed to support the hard bone or muscle tissues until they were entirely degraded after complete tissue regeneration. Therefore, with further optimization, they may be suitable for applications in cartilage tissue engineering.

FT-IR and XRD measurements were effective methods to explore the interaction among the components of composite scaffolds, and the chemical bonds of their internal components could be analyzed by detecting the changes in the functional groups and the crystal structure of the composite materials. Figure 5a shows the FT-IR spectra of the SA, BCNs, Alg-CS-GT and Alg/BCNs-CS-GT composite scaffolds. It can be seen that the BCNs exhibited characteristic peaks at 2939.30, 1457.62 and 1073.60 cm^−1^, respectively, for the asymmetric stretching vibration of –CH_2_– and the asymmetric deformation vibration of C–O–C on the pyranose ring [42]. In addition, the peaks appearing at 1733.21 and 863.58 cm^−1^ were assigned to the –COOH bending vibration and the C–O stretching vibration absorption peak, which implied the successful oxidation of anionic sulfate half-ester groups on the pyranose ring by hydrogen peroxide [43]. By contrast, it was found that the –OH stretching vibration absorption peaks of the Alg-CS-GT and Alg/BCNs-CS-GT composite scaffolds at 3000–4000 cm^−1^ became much broader than those of the SA and BCNs, and they were accompanied by a red shift effect to a certain degree. Furthermore, the Alg/BCNs-CS-GT composite scaffolds mainly displayed the characteristic peaks of SA, which had a higher degree of red shift relative to BCNs, and the peaks at 1733.21 and 863.58 cm^−1^, which were assigned to the characteristic peaks of BCNs, disappeared, indicating that the BCNs were fully embedded in the alginate matrix, forming the intermolecular hydrogen bond between the BCNs and the alginate matrix.

In addition, the XRD patterns of the SA, BCNs, Alg-CS-GT and Alg/BCNs-CS-GT composite scaffolds were presented in Figure 5b. Due to the hydrated crystalline structure formed by the ions’ cross-linking of alginate, the Alg-CS-GT composite scaffolds indicated two weak peaks at 2θ = 16.2° and 21.8° [11]. The BCNs showed their characteristic peaks at 2θ = 14.7°, 16.8° and 22.7°, corresponding to the (−110), (110) and (200) crystal planes [44]. Nevertheless, with the increase in BCNs content, the diffraction peaks of the Alg/BCNs-CS-GT composite scaffolds at 2θ = 16.2° and 2θ = 21.8° gradually shifted to 2θ = 14.7° and 22.7°, and their peak intensity gradually increased. However, compared with BCNs, their characteristic diffraction peak intensity was significantly reduced. As reported in the literature, the reduction in the intensity of the diffraction peak was attributed to the formation of hydrogen bonds between the BCNs and SA molecules in the Alg/BCNs-CS-GT composite scaffolds [45], which was consistent with the FT-IR analysis. At the same time, the coating and shielding effect of SA on BCNs also promoted the decrease in the characteristic diffraction peak intensity.

The swelling performance of the scaffold is an important indicator of its clinical application. Appropriate swellability not only increases the surface area of the scaffold but also promotes the transfer of nutrients, thereby promoting the adhesion and penetration of more cells on the scaffold [46]. However, excessive swelling does not only cause the relaxation of the scaffold material, affecting its mechanical integrity, but also exerts a pressure load on the surrounding tissue, which is not conducive to tissue repair and regeneration [1]. Therefore, controlling swelling is very important in tissue engineering applications. As shown in Figure 6a, the addition of BCNs enabled the swelling of the Alg/BCNs-CS-GT composite scaffolds to be effectively controlled. As the content of the BCNs increased, the swelling degree of the Alg/BCNs-CS-GT composite scaffolds gradually decreased. The result can be explained by the fact that the BCNs molecular chain contained a large number of –OH groups, which could form intermolecular hydrogen bonds with SA, and its tight fiber network structure and good barrier properties could also control the swelling of the Alg/BCNs-CS-GT composite scaffolds in PBS [15]. Therefore, the use of BCNs as the reinforcing agent of Alg/BCNs-CS-GT composite scaffolds not only improved the mechanical stability of the composite scaffolds, but could also effectively controlled the swelling of the composite scaffolds by controlling the content of BCNs.

The biodegradability of the scaffolds is another important indicator that needs to be considered in tissue engineering applications. Good scaffolds must have a suitable degradation rate to match the regeneration or repair of the tissue [4]. Figure 6b shows the biodegradation rate of the Alg/BCNs-CS-GT composite scaffolds in PBS containing 10,000 U/mL lysozyme at 37 °C. Over a short-term degradation period, such as 2 days or 6 days, the addition of BCNs did not significantly reduce the degradation rate of the Alg/BCNs-CS-GT composite scaffolds. This may be due to the protective effect of the composite scaffolds produced by the LBL assembly of polyelectrolytes, due to which their degradation rates were not more than 12%. Under the action of lysozyme, with the dissolution of the polyelectrolyte complex formed by CS and GT [46], and the destruction of ion crosslinking sites [13], the biodegradation rate of the Alg/BCNs-CS-GT composite scaffolds rose rapidly after about 10 days. At this time, the tight fiber network structure and good barrier properties of BCNs, and their formation of intermolecular hydrogen bonds with alginate, could effectively inhibit the degradation of the composite scaffolds and reduce their degradation rate. Therefore, when the amount of BCNs was higher than 0.4% (*w*/*v*), the biodegradation rate of the Alg/BCNs-CS-GT composite scaffolds was significantly lower than that of the Alg-CS-GT composite scaffolds.

In vitro biomineralization of Alg-CS-GT and Alg/BCNs-CS-GT composite scaffolds was examined in SBF solution to evaluate the apatite formation ability on their surface. As shown in Figure 7, compared to the untreated composite scaffolds, the Alg-CS-GT and Alg/BCNs-CS-GT composite scaffolds both formed a layer of bone-like apatite on their surface, with diameters of 1.2–2.5 μm, after being soaked in SBF solution for 14 days. Moreover, the surface of the Alg/BCNs-CS-GT composite scaffolds revealed more bone-like apatite than that of the Alg-CS-GT composite scaffolds, which proved that the addition of BCNs increased the efficiency of the mineralization process. EDX analysis was conducted to determine the chemical composition of the apatite crystallites presented in the SEM images. In Figure 7c, f, the Ca and P, the main elements of apatite, can be found on the surface of Alg-CS-GT and Alg/BCNs-CS-GT composite scaffolds, which could be applied to calculate the Ca/P ratio [47]. Based on the EDX analysis, the Ca/P ratios of the Alg-CS-GT and Alg/BCNs-CS-GT composite scaffolds are presented in Appendix A; these values are close to the value defined for hydroxyapatite (Ca/P = 1.67), indicating that the Alg-CS-GT and Alg/BCNs-CS-GT composite scaffolds both possessed apatite formation ability, which has relevance for biomedical applications [48].

.

### 2.4. In Vitro Cytotoxicity of Alg/BCNs-CS-GT Composite Scaffolds

The representative optical micrographs of osteosarcoma MG-63 cells and osteoblastic MC3T3-E1 cells on TCP, after they were seeded and cultured for 2 days, are shown in Figure 8. It can be observed that both the MG63 cells and MC3T3-E1 cells showed high growth activity within 2 days. Their morphologies of the Alg/BCNs-CS-GT composite scaffolds are shown in Figure 9 and Figure 10, respectively.

The SEM images of the composite scaffolds without cells after incubation for 2 days are shown in Appendix A. By contrast, due to the deposition of GT on the outer surface of the composite scaffolds, all of the composite scaffolds displayed cell morphologies. MG63 cells on the Alg/BCNs-CS-GT composite scaffolds presented a spherical particle morphology after fixation with glutaraldehyde. When the added amount of BCNs exceeded 0.2% (*w*/*v*), the MG63 cells showed better dispersibility on the composite scaffolds, indicating that BCNs could promote the distribution of MG63 cells on the composite scaffolds, which may be related to their porous fiber network, which can fully mimic the extracellular matrix [49]. However, the MC3T3-E1 cells showed irregular morphology due to their different degrees of spreading on the Alg/BCNs-CS-GT composite scaffolds. The MC3T3-E1 cells mainly presented a spherical shape on the Alg-CS-GT composite scaffolds, and it was difficult to observe their pseudopods. With the increase in BCNs content, the pseudopods of cells gradually spread on the Alg/BCNs-CS-GT composite scaffolds, which also indicated that BCNs promoted the adhesion and spreading of MC3T3-E1 cells on the composite scaffolds.

After 2 days and 5 days of incubation, the proliferation viability of the MG63 cells and MC3T3-E1 cells on the Alg/BCNs-CS-GT composite scaffolds was tested using the CCK-8 assay kit. As shown in Figure 11, the cells showed good proliferative capacity on the Alg-CS-GT and Alg/BCNs-CS-GT composite scaffolds, and their proliferation activities were higher than those of the control group, indicating that the cells were able to grow in the 3D-porous structure of the Alg-CS-GT and Alg/BCNs-CS-GT composite scaffolds. By comparison, the MG63 cells and MC3T3-E1 cells both exhibited similar proliferation trends on different types of scaffolds. With the increase in BCNs content, the proliferation activity of the cells first increased and then decreased. When the amount of BCNs was 0.2%~0.3% (*w*/*v*), the cells showed a significant proliferation effect on the composite scaffolds. In particular, when the amount of BCNs was 0.2% (*w*/*v*), the proliferation ability of cells on the composite scaffolds was significantly higher than that of the blank control group. The results may be related to the pore size of the composite scaffolds. As the BCNs content increased, the pore structure of the Alg/BCNs-CS-GT composite scaffolds decreased. It can be accepted that overly small pore structures were not conducive to cell osmotic growth and the transfer of metabolic waste and nutrients [37], thus resulting in lower proliferation viability, which was even lower than the viability of cells on the Alg-CS-GT composite scaffolds. As a result, based on the above analyses, the optimum BCNs content for the Alg/BCNs-CS-GT composite scaffolds was 0.2% (*w*/*v*).

## 3. Materials and Methods

### 3.1. Materials

Sodium alginate (SA, M_W_ = 432 kDa), positively-charged chitosan (CS, M_W_ = 860 kDa, deacetylation degree ≥ 95%), negatively-charged type B gelatin (GT) obtained by alkaline treatment of collagen, hydroxyapatite (HAP) and d-glucono-δ-lactone (GDL) were purchased from Aladdin Chemical Reagent Co., Ltd., Shanghai, China. Human osteosarcoma MG-63 cells and mouse osteoblastic MC3T3-E1 cells were purchased from the cell bank of the Chinese Academy of Sciences, Shanghai, China. DMEM medium, MEM-α medium, fetal bovine serum, trypsin (0.25% trypsin-EDTA), penicillin and streptomycin were purchased from Gibco, Thermo Fisher Scientific, USA. The Cell Counting Kit-8 (CCK-8) was obtained from Dojindo Chemical Laboratories, Kumamoto, Japan. The Triton X-100 and alkaline phosphatase (ALP) assay kit were purchased from Biyuntian Biotechnology Co., Ltd., Shanghai, China. All chemical reagents were of biological grade and used without further purification. Pristine bacterial cellulose (BC) was produced by *Acetobacter xylinum* (CGMCC5173), obtained from the China General Microbiological Culture Collection Center, based on the previous method [50],

### 3.2. Synthesis of BCNs

The purified BC gelatinous membranes were crushed into small pieces and further homogenized with a high shear homogenizer to obtain uniform BC aqueous suspension. BCNs were prepared by sulfuric acid hydrolysis of the BC aqueous suspension according to a previous method, but with some modifications [51,52]. Approximately 10 g of BC was dispersed in 200 mL of 50 wt% sulfuric acid under vigorous mechanical stirring. The hydrolysis was performed at 45 °C for 3 h, and the mixture was diluted five-fold to quench the hydrolysis reaction. Then, 15 mL of 30 wt% hydrogen peroxide was added to bleach and oxidize BCNs. Afterwards, the resultant suspension was centrifuged at 9000 rpm for 15 min to separate the crystals, which were washed and treated ultrasonically for 15 min to eliminate excess acid. Finally, the precipitate was further dialyzed against deionized water for 8 d using a dialyzing membrane with a molecular weight cutoff of 3500 to remove residual sulfuric acid as well as other low-molecular weight impurities. The resultant BCNs suspension was stored in a refrigerator for further use.

### 3.3. Fabrication of Alg/BCNs-CS-GT Composite Scaffolds

For the preparation of Alg/BCNs-CS-GT composite scaffolds, the incorporation of BCNs as the reinforcement in the alginate matrix and subsequent LBL assembly for the surface modification of alginate hydrogel was carried out through the alternate electrostatic deposition of positively-charged CS and negatively-charged GT, as illustrated in Scheme 1. Briefly, a certain amount of BCNs and a certain amount of HAP were ultrasonically dispersed in 2% (*w*/*v*) SA solution and stirred vigorously until a homogenous solution was formed. Then, a certain amount of GDL was added to this solution with vigorous stirring to initiate gelation. In our study, HAP-GDL complex with the molar ratio of 1:10 was used as the cross-linking system, and the molar ratio of Ca^2+^ from HAP and carboxyl from SA was fixed at 0.18. After stirring for 3 min, the resultant mixture was cast in a 12-well tissue culture plate and refrigerated at 4 °C for 24 h. After freeze drying, the dried composite hydrogel was alternately dipped into a 1% (*w*/*v*) CS solution containing 0.01 mol/L CaCl_2_ and 2% (*w*/*v*) GT solution containing 0.01 mol/L CaCl_2_ for 30 min and then rinsed by deionized water to remove any unbound ingredient. Afterwards, the composite scaffolds were incubated in the EDC/NHS (10 mM/10 mM) blend solutions overnight, and then rinsed by deionized water. Finally, the wet composite scaffolds were further freeze-dried to obtain the Alg/BCNs-CS-GT composite scaffolds. During the preparation of Alg/BCNs-CS-GT composite scaffolds, 0.1%, 0.2%, 0.3%, 0.4% and 0.5% (*w*/*v*) BCNs were, respectively, incorporated into the SA solution, and the corresponding composite scaffolds were coded as Alg/0.1%BCNs-CS-GT, Alg/0.2%BCNs-CS-GT, Alg/0.3%BCNs-CS-GT, Alg/0.4%BCNs-CS-GT and Alg/0.5%BCNs-CS-GT. For comparison, Alg-CS-GT composite scaffolds with the absence of BCNs were also prepared by the same method.

### 3.4. Characterization

The morphologies of samples were observed by scanning electron microscopy (SEM) with energy-dispersive X-ray (EDX) and transmission electron microscopy (TEM). SEM observation was executed by a Hitachi S-3000N scanning electron microscope (Japan) after fixing the samples on a brass holder and coating them with gold. TEM observation was performed using a JEM 2100 TEM (JEOL Co., Tokyo, Japan) at an acceleration voltage of 200 kV. TEM images of samples were obtained by placing a few drops of the aqueous dispersions of the samples on a carbon-coated copper grid, and evaporating the solvent prior to observation. The porosity of the composite scaffold was measured using the liquid displacement method. The absolute ethanol was selected as the displacement liquid to calculate the porosity with the following formula (Equation (1)) [40]:
(1)$$P\;=\;\frac{W_{2}\;-\;W_{1}}{\rho_{0}V}\;\times \;100\%$$
where *W*_1_ and *W*_2_ are the weight of the scaffold before and after immersion in absolute ethanol, respectively, *ρ*_0_ is the density of absolute ethanol and *V* is the volume of the composite scaffold. Three parallel experiments were carried out for every scaffold and the mean value was taken.

The size and zeta potential of samples were determined by using a Malvern Nano-ZS90 Zetasizer (UK) at a scattering angle of 90 °C at 25 °C, employing an (He-Ne) argon laser (λ = 633 nm). The FT-IR measurements were conducted with the Fourier transform infrared spectrophotometer (Nicolet-6700 FT-IR, Thermo Scientific, Waltham, MA, USA). Before the measurement, a small amount of dried sample and KBr were mixed, ground and compressed into disks for the test. The XRD pattern of the examples was examined using a Bruker AXS/D8 Advance X-ray diffractometer system (Karlsruhe, Germany) with Cu-Kα radiation (λ = 0.154 nm). The XRD was operated at 40 kV and 100 mA in a step scan mode. The scanning speed was 0.025 °/s. XRD measurements were performed over a 2θ range of 5–60°. The mechanical properties of the composite scaffolds were tested by a microcomputer-controlled electronic universal material testing machine (CTM8050, Xieqiang Instrument Manufacturing Co., Ltd., Shanghai, China). The samples were cut into circular disks with a diameter of 20 mm and a height of 10 mm for mechanical testing. The compressive tests were conducted in the controlled strain rate mode with a crosshead speed of 5 mm/min. The test stops when the sample is crushed or its strain is greater than 60%. On the stress–strain curve recorded by the instrument, the highest linear point is the compressive strength of the sample. Each group of samples was tested in parallel 5 times to take the average value.

### 3.5. In Vitro Swelling, Biodegradation and Biomineralization Studies

The medium uptake ability of the composite scaffolds was detected by immersing dry scaffolds, which were weighed in PBS at 37 °C, for a time period of 24 h. The wet weight of the scaffold was recorded at different time intervals after removal of excessive surface water by filter paper. Three parallel experiments were carried out for every scaffold and the mean value was taken. The swelling ratio of the composite scaffolds was calculated based on the weight difference between the dried and swollen composite scaffolds using the following equation (Equation (2)) [40,46]:
(2)$$\mathrm{Swelling}\;\mathrm{ratio}\;=\;\frac{\mathrm{wet}\;\mathrm{weight}\;-\;\mathrm{dry}\;\mathrm{weight}}{\mathrm{dry}\;\mathrm{weight}}$$

The in vitro biodegradation of the composite scaffolds under physiological conditions was tested by immersing dry scaffolds, whose initial weights were recorded, in PBS medium containing 10,000 U/mL lysozyme at 37 °C for a time period of 2, 6, 10 and 14 days. At the specific time intervals mentioned above, the scaffolds were removed from the solution, washed with deionized water, and dried. Three parallel experiments were carried out for each scaffold and the mean value was taken. The biodegradation ratio of the composite scaffolds could be calculated by the following formula (Equation (3)) [40,46]:
(3)$$\mathrm{Biodeg}\;\mathrm{radation}\;\mathrm{ratio}\;=\;\frac{\mathrm{initial}\;\mathrm{weight}\;-\;\mathrm{dry}\;\mathrm{weight}}{\mathrm{initial}\;\mathrm{weight}}\;\times \;100\%$$

The biomineralization study was conducted based on a previous method by immersing the composite scaffolds in simulated body fluid (SBF) at 37 °C for 14 days [46,53]. A quantity of 100 mL of SBF solution, which mimics human body fluid and contains all necessary minerals, was prepared through the use of 0.655 g of NaCl, 0.227 g of NaHCO_3_, 0.037 g of KCl, 0.018 g of Na_2_HPO_4_, 0.031 g of MgCl_2_·6H_2_O, and a certain volume of 1.0 mol/L hydrochloric acid solution, which adjusted the pH to 7.4. About 1.0 g of the composite scaffolds was incubated in 100 mL SBF solution at 37 °C. After incubation for 14 days, the composite scaffolds were removed and washed three times with deionized water to remove any excess minerals. After the composite scaffolds were finally lyophilized, the apatite layer formation on the scaffolds was analyzed by SEM observation and energy-dispersive X-ray (EDX).

### 3.6. Cytotoxicity Studies

#### 3.6.1. Cell Culture and Seeding

Human osteosarcoma MG-63 cells and osteoblastic MC3T3-E1 cells that could mimic the growth process of osteoblasts were chosen as the model cells to detect the cytocompatibility of the composite scaffolds. The MG-63 cells and MC3T3-E1 cells were, respectively, cultured with the culture medium that was composed of 90% DMEM or 90% MEM-α, 10% fetal bovine serum, 100 U/mL penicillin and 100 μg/mL streptomycin. The composite scaffolds, with a bottom diameter of 14 mm and a height of 2 mm for cytocompatibility experiments, were prepared using a 24-well tissue culture plate as a mold according to the method in Section 2.3. The composite scaffolds were sterilized by cobalt 60 radiation (with an irradiation intensity of 8 kGy), and then they were placed into the 24-well tissue culture plates. The MG-63 cells and MC3T3-E1 cells were seeded on the composite scaffolds in the 24-well tissue culture plates at a density of 5 × 10^4^ per well, while the same cells were seeded on the tissue culture plates without scaffold material as a blank control. Afterwards, the culture medium was replenished to make the total amount of medium per well reach up to 500 μL. These tissue culture plates were then transferred to an incubator containing 5% CO_2_, 95% air and 100% relative humidity at 37 °C and their culture media were replaced every 2 days.

#### 3.6.2. Cell Attachment

The adhesion and spreading of the cells on the composite scaffolds was examined based on a previous method [2]. After incubation for 2 d, the MG-63 cells and MC3T3-E1 cells adhering on the composite scaffolds were fixed through chemical cross-linking with glutaraldehyde, washed with distilled water, and dehydrated with gradient concentrations of ethanol and tertiary butanol. Finally, after freeze drying, the adhesion and distribution of the MG-63 cells and MC3T3-E1 cells on the surface of the composite scaffolds could be observed by scanning electron microscope.

#### 3.6.3. Cell Proliferation

The proliferation activity of the cells on the composite scaffolds was evaluated using the Cell Counting Kit-8 (CCK-8) assay. After adding a certain amount of CCK-8 reagent, the absorbance value (OD) of the cell medium on the composite scaffolds was proportional to the number of living cells, so the proliferation activity of the MG-63 cells and MC3T3-E1 cells on the composite scaffolds could be judged by comparing the OD values. After the MG-63 cells and MC3T3-E1 cells were seeded on the composite scaffolds and, respectively, cultured for 2 days and 5 days, 30 μL of CCK-8 reagent was added to 500 μL of medium in each well in the 24-well culture plates, and then they were placed in the incubator at 37 °C for 4 h. Finally, 100 μL of solution from each well was transferred to a 96-well plate. Its OD value was determined by an X-mark microplate reader (Bio-Rad, Hercules, CA, USA) at a wavelength of 450 nm.

### 3.7. Statistical Analysis

All data were presented as means ± standard deviation (SD). Single-factor ANOVA was performed by SPSS software to analyze the variables, and a value of *p* < 0.05 was considered statistically significant.

## 4. Conclusions

In summary, BCNs with small particle size and good interfacial properties were prepared by sulfuric acid hydrolysis from BC. Furthermore, Alg/BCNs-CS-GT composite scaffolds were successfully constructed by endogenous cross-linking of the HAP-GDL complex and the LBL assembly of polyelectrolytes, using BCNs as the reinforcing agent. The effects of various contents of BCNs on the morphology, mechanical properties, swelling properties, degradation properties and cell compatibility of the composite scaffolds were investigated. The characterization of composite scaffolds indicated that the Alg/BCNs-CS-GT composite scaffolds had a regular 3D morphology and a well-developed pore structure. The content of BCNs had a significant effect on the pore size of the Alg/BCNs-CS-GT composite scaffolds. With the increase in BCNs content, the pore size of the Alg/BCNs-CS-GT composite scaffolds was gradually reduced from 200 μm to 70 μm. FT-IR and XRD analysis showed that the BCNs were fully embedded in the alginate matrix, and this produced an intermolecular hydrogen bond with SA. Moreover, the addition of BCNs effectively controlled the swelling and biodegradation of the Alg/BCNs-CS-GT composite scaffolds. Finally, the in vitro cytotoxicity studies of the Alg/BCNs-CS-GT composite scaffolds showed that the porous fiber network of BCNs could fully mimic the extracellular matrix structure, which promoted the adhesion and spreading of MG63 cells and MC3T3-E1 cells on the composite scaffolds. The MG63 cells and MC3T3-E1 cells showed good proliferative viability on the Alg/BCNs-CS-GT composite scaffolds, indicating that the cells could grow in the 3D-porous structure of the composite scaffolds. According to the BCNs’ effects, the optimum BCNs content for the Alg/BCNs-CS-GT composite scaffolds was 0.2% (*w*/*v*). Consequently, the Alg/BCNs-CS-GT composite scaffolds possessed a regular 3D morphology, a well-developed pore structure, controlled swelling and biodegradation behavior, and good cytocompatibility, and could therefore realize the cells’ adhesion, proliferation and differentiation on their surfaces, making them ideal candidates for biomedical applications in tissue engineering.

## Data Availability

The data presented in this study are available on request from the corresponding author.

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
