# Peer review of "Fabrication and Evaluation of Alginate/Bacterial Cellulose Nanocrystals–Chitosan–Gelatin Composite Scaffolds"

_molecules, 2021, doi:10.3390/molecules26165003_

Round 1

Reviewer 1 Report

  1. Title is long and difficult to follow. Please rephrase.
  2. English must be thoroughly checked, and many phrases should be re-written. You may find just a few examples below.
    1. “It is common knowledge that pure alginate hydrogel is more likely to undergo lower and unstable mechanical strength, the lack of cell recognition sites, and the extensive swelling and uncontrolled degradation, which is incapable of satisfying the demands of the ideal scaffold.”
    2. “Furthermore, the cross-linking structures produced by alginate and divalent Ca2+ that are easily destroyed in biological buffers containing chelators or monovaleat electrolytes, resulting in the extensive swelling and uncontrolled degradation [13].”
    3. “After it was freeze-dried, its weight (Wt) was re-weighed.”
    4. “Due to the small particle size and good interfacial property of BCNs, they are uniformly dispersed in the alginate matrix, so that the wet Alg/BCNs-CS-GT composite hydrogel revealed uniform light blue.”
    5. Etc
  3. Please provide bibliographic references for equations (1) and (2), respectively.
  4. Scheme 1 should be re-done, as it is not representative for the fabrication process.
  5. Please indicate the bibliographic reference describing the protocol used for the biomineralization tests. (volume of SBF / sample, static/dynamic conditions, was the SBF changed during the 14 days of tests, was anti-bactericide added in the SBF?...)
  6. “composite hydrogel scaffolds” – this expression, found throughout the manuscript, is difficult do follow. I suggest replacing it with “composite scaffolds”, “composite hydrogels”, “composites”, “composite materials”, etc.
  7. Please provide the stress-strain curves either in the text or as supplementary materials
  8. @ Figure 6: please redo the swelling ratio-time graph and represent the initial moment of the test (at time zero – swelling is zero). Moreover, the test should be performed until the samples reach equilibrium (at least three consecutive measurements taken at relevant datapoints indicating the same weight of the samples)
  9. Please provide the EDX report showing the values for Ca and P, either as inset in Figure 8 or as supplementary material. The EDX provided doesn’t seem to lead to a report Ca/P of 1.67

Reviewer 2 Report

This work investigated the potential application of fabricate alginate/bacterial cellulose nanocrystals-chitosan-gelatin (Alg/BCNs-CS-GT) composite hydrogel scaffolds for bone tissue engineering. The authors performed numerous physical property assessments, including biological evaluations.

  1. The authors state that the study of this composite is to compensate for the lack of mechanical strength in the alginate hydrogel, and while the compressive strength is indeed significantly higher by supplementing with BCNs, it is still not double, and not at all at the level required by scaffolding. The authors need to include some comments about this point.
  2.  Figure 7 shows the SEM images of Alg-CS-GT composite hydrogel scaffolds and Alg/0.3%BCNs-CS-GT composite hydrogel scaffolds soaked in SBF for 14 days. First, untreated SEM images of both scaffolds should also be shown. Furthermore, the authors state that the Alg/0.3%BCNs-CS-GT composite hydrogel scaffold has a higher ability to induce the deposition of bone-like apatite. The reviewer cannot make such a judgment from the SEM images. Please provide the evidence for this claim.
  3. The authors show SEM images of MG63 and MC3T3-E1 cells on a composite hydrogel scaffold in Figures 9 and 10. Since both are adherent cells, their spherical shape is probably not indicative of a healthy state; please show the basis for their assessment as having "good adhesion". Also, please add the SEM image of the composite hydrogel scaffold under the same conditions without cells for comparison.
  4. The reviewer has questions about the cell proliferation test. First, please describe the sample size of the Alg/BCNs-CS-GT composite hydrogel used in this test. At least the sample shown in Figure 3(b) should not fit into a 24-well plate.
  5. The linearity of the microplate reader used by the authors ranges from 0 to 3.0 OD (405 nm) according to the manufacturer's information. The authors show a value higher than that on day 2, but have they verified that this value is linear to the cell count? In general, it is considered that the reaction is already saturated under the measurement conditions on day 2 using CCK-8, and it is not possible to make a quantitative evaluation including data on day 5. The reviewer deems that re-experimentation is necessary. 

Reviewer 3 Report

This paper is about the preparation and evaluation of alginate-chitosan-gelatin composite based bacterial cellulose nanocrystals composite hydrogel scaffolds. The topic of this manuscript is interesting and up to date and it addresses an important issue regarding developing scaffold materials for tissue engineering.

The manuscript itself is well drafted and scientifically sound. The evaluations of results are sufficient, clear and contain sufficient information. The Figures are also clear and demonstrative.

My comments on this manuscript as follows:

The Authors should supplement more recent references related to this research area.

Reviewer 4 Report

A very interesting article. There is no abbreviation at the end of the article. The person reading it may understand the signs better. Have the authors tried to determine how many individual polysaccharides are in the implant? What is the total porosity of this implant? What is the stability of the SBF implant or its degradation time?

Round 2

Reviewer 1 Report

Following the first revision, the authors made some changes in the manuscript, but it still has several flaws. Firstly, the manuscript should be checked for grammar errors and faulty phrasing. The confusing phrasing makes the manuscript hard to follow.

Furthermore, given the complexity of both composition and fabrication method, there are many aspects not taken into consideration by the authors. For example, in the case of the mechanical properties, the compressive strength increases with the addition of BCN, but there is also the matter of strain value alt which fracture occurs. Judging from the strain-stress curves, the Alg/ 0.5% BCNs-CS-GT sample fractures around 0.22 MPa (not at 0.314 as reported!) at a strain at 20%. The Alg/ 0.4% BCNs-CS-GT has a similar compressive strength at almost half the strain value. The value of the strain should also be taken into consideration. Not only the addition of BCN influences the compressive strength value, but also the physical bonds established between the components of the system.

In addition, the introduction of this study is based on the hypothesis that these materials have potential for bone tissue engineering, but the mechanical properties demonstrate otherwise. The authors mention at the end of section 3.3 that the materials “were more suitable for the application in cartilage tissue engineering.”. However, this last statement is superficial, since the cartilage tissue must withstand large deformations without braking.

The porosity of the scaffolds should be related to their swelling ability. However, the 0.3% BCN sample exhibits higher porosity when compared to 0.4 and 0.5% BCN samples. The authors should exploit other measurements methods and explain why the porosity decreases when the swelling increased.

Table S1, provided by the authors does not represent the EDX report. As much as I can see if figure 7 g & h, the Ca/P ratio is around  2.5 for Alg-CS-GT and 1.3 for Alg/0.3% BCNs-CS-GT

Reviewer 2 Report

The reviewer accepts to publish this paper.

Author Response

Thank you for your carefulness, conscientious, and your kind attention again.